# Overview on the Different Patterns of Tumor Vascularization

**DOI:** 10.3390/cells10030639

**Published:** 2021-03-13

**Authors:** Domenico Ribatti, Francesco Pezzella

**Affiliations:** 1Department of Basic Medical Sciences, Neurosciences and Sensory Organs, University of Bari Medical School, 70124 Bari, Italy; 2Nuffield Division of Laboratory Science, Radcliffe Department of Medicine, University of Oxford, John Radcliffe Hospital, Oxford OX39DU, UK

**Keywords:** angiogenesis, tumor growth, vascular co-option, vasculogenic mimicry

## Abstract

Angiogenesis is a crucial event in the physiological processes of embryogenesis and wound healing. During malignant transformation, dysregulation of angiogenesis leads to the formation of a vascular network of tumor-associated capillaries promoting survival and proliferation of the tumor cells. Starting with the hypothesis formulated by Judah Folkman that tumor growth is angiogenesis-dependent, this area of research has a solid scientific foundation and inhibition of angiogenesis is a major area of therapeutic development for the treatment of cancer. Over this period numerous authors published data of vascularization of tumors, which attributed the cause of neo-vascularization to various factors including inflammation, release of angiogenic cytokines, vasodilatation, and increased tumor metabolism. More recently, it has been demonstrated that tumor vasculature is not necessarily derived by endothelial cell proliferation and sprouting of new capillaries, but alternative vascularization mechanisms have been described, namely vascular co-option and vasculogenic mimicry. In this article, we have analyzed the mechanisms involved in tumor vascularization in association with classical angiogenesis, including post-natal vasculogenesis, intussusceptive microvascular growth, vascular co-option, and vasculogenic mimicry. We have also discussed the role of these alternative mechanism in resistance to anti-angiogenic therapy and potential therapeutic approaches to overcome resistance.

## 1. Background

In 1787, the concept of angiogenesis was introduced by John Hunter, the founder of scientific surgery, describing the process through which new blood vessels arise from preexisting ones, [1,2,3]. Hunter stated that vessels “*would appear to have more powers of perfecting themselves, when injured, that any part of the body; for their use is almost immediate and constant, and it is they which perform the operation of restoration on the other parts, therefore they themselves must be perfect*” [4]. The actual term “angiogenesis” was instead introduced by Flint in 1900, when he described the vascularization of the adrenal gland referring to the work of previous scientists related to the new formation of capillaries from pre-existing ones [5].

An awareness that cancers contain blood vessels goes back to the dawn of medicine. Indeed, according to legend, Paul of Aegina reported in the 7th century AD that cancer was so called because this disease “*has the veins stretched on all sides as the animal the crab has its feet, whence it derives its name*” [6]. However, the first historical description of blood vessels related to cancer is commonly accredited to Galen of Pergamon (AD 129-c. 200/c. 216) in his treaty *On abnormal swellings*: “*Whenever black bile attacks the flesh, because it is corrosive, it eats into the surrounding skin producing an ulcer. If it arises with less intensity, it causes a cancerous swelling without ulceration. As I have said before, the veins are distended by black bile to a greater extent than in inflammatory swelling, regardless of the colors that they appear to be*” [7].

However, tumor vascularization was first studied systematically only at the beginning of the 20th century by Goldman [8], who visualized the vascular networks in human and animal tumors by injecting India ink and bismuth-in-oil- in the vasculature, and described the morphological characteristics of tumor blood vessels as follows: “*The normal blood vessels of the organs in which the tumor is developing are disturbed by chaotic growth, there is a dilatation and spiraling of the affected vessels, marked capillary budding and new vessel formation, particularly at the advancing border*”.

The conclusion that tumor growth relies on sprouting angiogenesis [9] was largely based on in vitro experiments and animal models where the experiments were conducted in relatively avascular sites, such as the cornea and the ear of the rabbit. Despite this non-physiological condition (most cancers arise in well vascularized tissues) these experiments were regarded as a convincing proof of concept.

Clark et al. [10,11] studied the tumor blood vessels in vivo through implantation of transparent glass-windowed chambers in a rabbit’s ear. In 1939, Ide et al. [12] suggested that tumors release specific factors capable of stimulating the growth of blood vessels. In 1943–45, Algire and co-workers [13,14,15] used a transparent chamber implanted in a cat’s skin to study the vasoproliferative reaction secondary to the implantation of neoplastic tissues, and showed that the response induced by tumor tissues was more substantial and earlier than that induced by normal tissues. They also quantified the number of blood vessels and demonstrated that the tumor growth is closely related to the development of an intrinsic vascular network [13,14,15]. An Italian pathologist, Rondoni, suggested that a tumor acts both angioplastically and angiotactically, promoting new vessel formation and attracting pre-existing capillaries [16,17]. In 1956, Merwin and Algire [13] found that the vasoproliferative response of normal or neoplastic tissues transplanted into muscle was not significantly different as concerns the time of onset of new blood vessels, and it was stronger when the implantation was performed in a resection area. Moreover, the intensity of the response seemed to be influenced by the distance between the implant and the host’s vessels. Greenblatt and Shubik [18] implanted Millipore chambers into a hamster’s cheek pouch and placed tumor fragments around them. In a few days, the growing tumor mass engulfed the whole chamber, whose pores were permeable to the tumor interstitial fluid, but not to the tumor cells. New blood vessels were formed very likely through the release of a diffusible factor that could pass through the pores.

## 2. Isolation of the First Tumor Angiogenic Factor

The observation that new vessels are formed within tumors lead to investigate the biological basis of angiogenesis. In 1971, Folkman and co-workers isolated the first angiogenic factor, namely “tumor angiogenesis factor” (TAF) from the homogenate of a Walker 256 carcinoma. TAF has since been extracted from several tumor cell lines, and several low molecular weight angiogenic factors have been isolated, again from the Walker 256 carcinoma. These factors induced a vasoproliferative response in vivo when tested on rabbit cornea or chick chorioallantoic membrane (CAM), and in vitro on cultured endothelial cells [19,20,21].

In 1971, Folkman published a hypothesis that, as tumor growth [22] is angiogenesis-dependent, inhibition of angiogenesis could be therapeutic. The hypothesis predicted that tumors would be unable to grow beyond a microscopic size of 1 to 2 mm^3^ without continuous recruitment of new capillary blood vessels. This paper also introduced the term anti-angiogenesis to mean the prevention of new vessel sprout from being recruited by a tumor.

The 1980s saw the discovery of other molecules that mediate angiogenesis. Heparin-affinity chromatography was employed to purify basic fibroblast growth factor or fibroblast growth factors-2 (FGF-2) (Maciag et al. [23]; Shing et al. [24]) and vascular endothelial growth factor (VEGF) [25,26]. In 1989, Ferrara and Henzel and Plouet et al. [25,26] independently reported the purification and sequencing of an endothelial cell-specific mitogen, which they respectively called VEGF and vascular permeability factor (VPF). The subsequent molecular cloning of VEGF and VPF [27,28,29] revealed that both activities are embodied in the same molecule.

## 3. The Avascular and Vascular Phases of Solid Tumor Growth

On the basis of the experiments discussed above, the growth of solid tumor growth was described to consists of an avascular and a subsequent vascular phase. Acquisition of angiogenic capability can be seen as an expression of progression from neoplastic transformation to tumor growth and metastasis. The earliest evidence of the existence of the two phases was obtained in 1963 by Folkman et al. [30], who perfused the lobe of a thyroid gland with plasma and inoculated a suspension of melanoma B16 tumor cells through the perfusion fluid. These cells grew into small, clearly visible black nodules. The nodules did not exceed 1 mm in diameter and did not connect with the host’s vascular network. Re-implanted nodules, on the other hand, equipped themselves with a vascular network and grew very rapidly. The conclusion was thus that the absence of vascularization limits the growth of solid tumors.

Folkman and collaborators provide further evidence for the dependence of tumor growth on neovascularization: tumor growth in the avascular cornea proceeds slowly at a linear rate, but after vascularization, tumor growth is exponential [31,32]. Tumors suspended in the aqueous fluid of the anterior chamber of the rabbit eye and observed for a period up to 6 weeks remain viable, avascular, of limited size (less than 1 mm^3^) and contain a population of viable and mitotically active tumor cells. After implantation contiguous to the iris, which has abundant blood vessels, the tumors induced neovascularization and grow rapidly, reaching 16,000 times the original size within two weeks After oneweek, new blood vessels had invaded the cornea starting from the edge closer to the site of implantation and developed in that direction at 0.2 mm and then about 1 mm/day. Once the vessels reached the tumor, it grew very rapidly to permeate the entire globe within four weeks. (Figure 1).

Gullino’s group [33,34,35] observed that experimental breast cancer in rat and mouse gave rise to marked breast angiogenic activity that was lacking in adult gland. Moreover, pre-neoplastic lesions also induce a strong vasoproliferative response long before any morphological sign of neoplastic transformation can be observed. Resting adult rodent mammary gland has limited, if any angiogenic capacity, whereas this is consistently acquired by mammary carcinomas. Lesions with a high frequency of neoplastic transformation induce angiogenesis at a much higher rate than those with a low frequency, and this elevated angiogenic capacity was observed long before any morphological sign of neoplastic transformation. Hyperplastic human mammary gland lesions behave in the same way and angiogenesis may thus be an early marker of neoplastic transformation.

The event allowing the transition from avascular to vascular phase was called “the angiogenic switch”: this mechanism [36] has been inaccessible to analysis until the development of an experimental model in which the large “T” oncogene is hybridized to the insulin promoter [37]. In this islet cell tumorigenesis (RIP1-TAG2 model), mice express the large T antigen (TAG) in all their islet cells at birth, and express the SV40 TAG under the control of the insulin gene promoter, which elicits the sequential development of tumors in the islets over a period of 12–14 weeks. Tumor development proceeds by stages during which about half the 400 islets hyper-proliferate, while about 25% of them subsequently acquire the ability to switch to angiogenesis. In this model, the pancreatic β cells become hyperplastic and progress to tumors via a reproducible and predictable multistep process [38], and the switch depends on increased production of one or more of the positive regulators of angiogenesis.

## 4. Tumor Angiogenesis in Human Tumor Progression as a Prognostic Indicator

Angiogenesis and production of angiogenic factors are fundamental for tumor progression in form of growth, invasion, and metastasis. New vessels promote growth by conveying oxygen and nutrients and removing catabolites. In some tumors, an oncogene is responsible for overexpression of a positive regulator of angiogenesis. For example, Ras oncogene upregulates expression of VEGF. In other tumor types, concomitant down-regulation of an endogenous negative regulator of angiogenesis is also required for the angiogenic switch. When the tumor suppressor gene TP53 is mutated or deleted, thrombospondin-1 (TSP-1), an endogenous angiogenesis inhibitor, is downregulated. Tumor angiogenesis can be potentiated by hypoxia, which activates a hypoxia-inducible factor (HIF-1)-binding sequence in the VEGF promoter, leading to transcription of VEGF mRNA, and increased production of VEGF protein. HIF-1 also increases the transcription of genes for platelet-derived endothelial growth factor (PDGF)-BB and nitric oxide synthetase (NOS). Tumor angiogenesis can also be potentiated by angiogenic proteins released by inflammatory cells present in the tumor microenvironment.

Numerous literature evidence has demonstrated a close relationship between tumor progression and angiogenesis, firstly in solid tumors. In 1994, for the first time, Vacca et al. demonstrated bone marrow angiogenesis in multiple myeloma and a high correlation between the extent of neovascularization and tumor plasma cell proliferation [39]. Ribatti el al. [40] firstly showed the bone marrow angiogenesis also in B-cell non-Hodgkin’s lymphomas and Perez-Atayde et al. [41] in acute lymphocytic leukemia.

Angiogenesis is a prognostic indicator for a wide variety of tumors. The first report appeared in 1988, when Srivastava et al. [42] found that the degree of histologic staining for vessels in melanoma patients was associated with a probability of metastasis. Weidner et al. [43] showed a direct correlation between microvascular density (MVD) and metastasis in human breast cancer. This finding has since been extended to different solid and hematological tumors. The effect of MVD as prognostic indicator has been however questioned by a meta-analysis on original data that failed to confirm the value of MVD as a prognostic marker in non-small cell lung carcinoma [44].

It is important to consider the relationship between MVD and intercapillary distance. Intercapillary distance is determined at the local level by the net balance between angiogenic and anti-angiogenic factors, as well as by the oxygen and nutrient consumption rates of tumor cells [45]. In turn, MVD is determined by intercapillary distance, which in a tumor is dictated by the thickness of the perivascular cuff of tumor cells. Tumors that have high rates of oxygen and nutrient consumption have small cuffs only two to three cells wide and have high vascular density [46]. On the contrary, tumors with low rates of oxygen consumption have relatively large cuff sizes and a relatively low vascular density [46]. This is an important parameter as the goal of an anti-angiogenic tumor therapy to reduce the intercapillary distance to a such a degree that it becomes rate-limiting for the tumor growth.

## 5. Tumors Can Growth without Inducing Angiogenesis

The histopathological analyses of growth patterns of different types of carcinoma in patient samples provided novel and interesting insights into the relation of tumor growth and vessels. In fact, independent teams have demonstrated that tumors, by adopting specific growth patterns, can in fact grow without the need to induce angiogenesis [47]. The growth of these tumors is achieved because they respect the architecture of the host stroma and thereby co-opt the pre-existing blood vessels as a means of tumor vascularisation [48,49,50,51]. Perhaps unsurprisingly, this non-angiogenic form of tumor growth was first observed in organs with a dense vascular network such as the lung, the liver and the brain. The phenomenon of non-angiogenic growth by vascular co-option (also commonly termed vessel co-option) has subsequently been described also in animal models [52,53,54] (Figure 2). Non-angiogenic tumors grow in the absence of angiogenesis by two main mechanisms: cancer cells infiltrating and occupying the normal tissues to exploit pre-existing vessels (vascular co-option or vessel co-option); the cancer cells themselves forms channels able to provide blood flow (the so called vasculogenic mimicry) [47].

As the scientific community has investigated the relationship between blood vessels and cancer for more than a century, it may be surprising that non-angiogenic tumor growth has been largely overlooked. But, as a matter of fact, it has not gone unnoticed. Indeed, numerous isolated observations have been published in the past that contrast with the theory that all tumors require ‘sustained angiogenesis.’ Reading this literature proved to be a very enlightening experience.

In 1988, Kolin and Koutoulakis [55], while investigating the role of arterial occlusion in some primary lung scar carcinomas, noticed that the neoplastic cells were “*often growing mainly in air spaces and preserving the pulmonary framework as their stroma*”. Although that paper was not primarily concerned with the issue of vessels in neoplastic growth, the authors highlight that “*As long as the proliferating lung cancer spreads through the air spaces without destroying the alveolar septa, it is nourished by the original alveolar capillaries, fed mainly from pulmonary circulation*.” In 1934, Willis described in his book *The Spread of Tumours in the Human Body* [56] peri-vascular growth of metastatic cells in the brain as well as “intra-alveolar” growth in the lung, two manifestations of non-angiogenic tumor growth (Figure 3). Regarding the lung metastases, he wrote that “*Intra alveolar growth of tumors in the lung is a characteristic and frequent mode of extension*” in which “*the plugs of growth occupying the air sacs are themselves avascular, the septal walls constituting the only stroma of the tumors*.” He also quoted several papers, including a case report from 1869 [57] and an even older one from 1861, which he accredits as the first description of the “intra-alveolar” growth pattern of tumors. In this paper the author described “a carcinoma nodule of the lungs” in which “the alveoli are shaped by cancerous masses confined between the elastic fibers of the lung tissue.” (Figure 4) [58]. In the paper of 1869, published in Transactions of the Pathological Society of London, a case report of “*transplantation of epithelial cancer from the trachea to the pulmonary tissue, probably by descent of cancer germs down the bronchial tubes*” was presented. The neoplasia was located in the esophagus, “*opposite the middle of the trachea*.” The only other localizations were deep into both lungs. The esophageal biopsy showed an epithelial malignancy with “bird’s nest” appearance: “*The little growths in the lung showed the same structure, yet even more beautifully, the “bird’s nest” capsules being here remarkably perfect and striking to the eye*” [57]. But it is the manuscript of Erichsen of 1861 [58], “Zwei falle von carcinosis acuta miliaris” (two cases of acute miliary carcinomatosis), that is the first to describe how, in patients with tumors in the lung, the neoplastic cells occupy the alveolar spaces but no new vessels can be seen. He also provided an iconographic representation to illustrate this point with a remarkable “Camera Lucida” drawing.

At the end of the 1980s in the book *Tumours of the Liver and Intrahepatic Bile Ducts* [59] a type of hepatocellular carcinoma is portrayed in which abnormal hepatocytes grow as trabecula and take the place of the normal hepatocytes. Interpreted at the time as “*probably hepatocellular carcinoma*” this malignancy has now been confirmed to be a primary liver tumor that grows in a non-angiogenic fashion by co-opting pre-existing sinusoidal vessels of liver, as subsequently illustrated [50]. Early in the 1960s, liver metastases with a similar peculiar growth pattern were described. The cancer cells form cell plates that resemble the liver cell plates by replacing the hepatocytes, and, doing so, they leave the stroma with the sinusoidal blood vessels intact. The authors stated that: “*Replacement is the most astonishing and the least comprehensible of the modes of behavior ascribed to metastatic carcinoma*” [60]. More recent studies have shown that this replacement growth pattern of liver metastases is indeed non-angiogenic [50], as the cancer cells co-opt the sinusoidal vasculature of the liver. Notably, nearly all human breast cancer liver metastases adopt this non-angiogenic replacement growth pattern [61].

However, what is perhaps most intriguing it is that vessel co-option has been extensively described, not only in dedicated research papers or in specialist monographs, but also in what are, arguably, the most successful and most popular textbooks of modern pathology. Indeed, in the 1974 edition of the book *Robin’s Pathological Basis of Disease* it is reported that, as far as lung metastases is concerned: “*Occasionally, the tumour seems to fill the alveolar spaces while preserving the intervening septa, and thus produces a solidification without destruction of the lung parenchyma, resembling superficially the exudative pattern of bacterial pneumonia*” [62]. Moreover, in Florey’s *General Pathology* (1962), when the issue of cancer vascularization is addressed [63], one can read that: “*…a tumour will supplement or replace the stroma by making use of pre-existing structures. For example, occasional tumours in the lung grow around the alveoli using the alveolar walls in place of stroma*.” These descriptions must have been read by countless medical students!

## 6. Vascular Co-Option

As we mentioned before, most of the non-angiogenic tumors grow by mean of vascular co-option. This definition was firstly used in 1999, when Holash et al. reported that some experimental tumors initially grow by exploiting the host vessels and initiate a pre-existing blood-vessel-dependent tumor growth [53]. These vessels then regress owing to apoptosis of the constituent endothelial cell, apparently mediated by angiopoietin- 2 (Ang-2). Angiogenesis occurs at the periphery of the growing tumor mass by cooperative interaction of VEGF and Ang-2. Therefore, these tumors eventually trigger angiogenesis [53]. The ability of cancer cells to co-opt the vessels initiates the process through few of them will intravasate, starting the metastatic process [64]. The blood vessels at the interface between tumor and surrounding tissue can be co-opted through the replacement of epithelial cells by tumor cells and/or the invasion of tumor cells into the stroma surrounding the blood vessels [65,66]. Vascular co-option is more frequently observed in tumors of densely vascularized organs, including brain, lung, liver, where the primary tumor cells co-opt the adjacent quiescent blood vessels of the hots tissue.

A significant greater number of patients with non-angiogenic vs angiogenic primary lung tumors developed distant lung metastases and vessel-co-option was associated with decreased overall survival [67,68]. In the first report on the growth pattern of breast cancer lung metastases, non-angiogenic growth was present in 19% of the lung metastases [69]. A 3D-tumor reconstruction showed that the blood vessel immunostaining resembled normal lung [70].

The studies of Barnhill and Lugassy have shown the extravascular migration of melanoma cells in brain metastasis model along the blood vessels of the tissue surrounding the tumor, constituting an alternative mechanism of tumor spreading instead of intravascular dissemination [71]. These authors introduced the terms “angiotropism” and “pericytic mimicry”, respectively, to indicate the recruitment and the migration of tumor cells along the basement membranes [72,73].

## 7. Vasculogenic Mimicry

Maniotis et al. [74] described a new model of formation of vascular channels by human melanoma cells and called it ‘‘vasculogenic mimicry’’ to emphasize the de novo generation of vascular channels without the participation of endothelial cells. This is the first study demonstrating the high plasticity of cancer cells to participate in the construction of new vessels in response to increased hypoxia. Indeed, the transendothelial differentiation of melanoma cells results in the formation of a chimeric vasculature composed of human melanoma and mouse endothelial cells. After restoring blood flow, the melanoma cells form a large tumor mass. This process occurs in the absence of endothelial cells and the tumor cells mimic the vascular endothelium.

A basement membrane stained with periodic acid Schiff (PAS) can be observed in these vascular channels, but endothelial cells were not identified by light microscopy, by transmission electron microscopy, or by using an immunohistochemical panel of endothelial cell markers [75]. Microarray gene chip analysis of a highly aggressive compared with poorly aggressive human cutaneous melanoma cell lines revealed a significant increase in the expression of laminin 5 and matrix metalloproteinases-1, -2, and -9 (MMP-1, MMP-2, MMP-9) and MT1-MMP in the highly aggressive cells [76], suggesting that increased expression of MMP-2 and MT1-MMP along with matrix deposition of laminin 5 are required for their mimicry.

Vasculogenic mimicry has been demonstrated in different human tumors, including renal cell carcinoma, breast cancer, ovarian cancer, primary gallbladder cancer, esophageal squamous cell carcinoma, mesothelioma, alveolar rhabdomyosarcoma, and hepatocellular carcinoma [75]. Vasculogenic mimicry can serve as a marker for tumor metastasis, a poor prognosis, worse survival, and a highest risk of cancer recurrence. In fact, the density of vasculogenic mimicry in poorly differentiated tumors is higher that of well-differentiated tumors.

## 8. Intussusceptive Angiogenesis

Next to the co-option and vascular mimicry another alternative way of tumor angiogenesis, so called intussusceptive angiogenesis or intussusceptive microvascular growth (IMG) has been reported. IMG is characterized by the insertion of newly formed columns of interstitial tissue structures called tissue pillars into the vascular lumen [77].

Even more, it seems that this complementary way of blood vessel formation could be switched one during tumor treatment with various antitumor and anti-angiogenic molecules. Both irradiation and anti-angiogenic therapy cause a switch from sprouting to IMG, representing an escape mechanism and accounting for the development of resistance, as well as rapid recovery, after cessation of therapy [78].

## 9. Concluding Remarks

Much research effort has been concentrated on the role of angiogenesis in cancer (Table 1) and inhibition of angiogenesis is a major area of therapeutic development for the treatment of tumors.

Two other mechanisms together with those previously described in this article are involved in the formation of tumor vessels, i.e., post-natal tumor vasculogenesis and endothelial prercursor cells (EPCs) activation [79,80]. Vasculogenesis may not be restricted to early embryogenesis, but may also contribute to tumor vascularization in postnatal life. This comes from observations that both endothelial cells and EPCs co-exist in the circulation and that blood vessels can emerge in an alternative way during tumor growth. EPCs may derive from the bone marrow and not yet be incorporated into the vessel wall. Several evidences suggest that EPCs constitute the preponderance of circulating bone marrow-derived endothelial lineage cells. In some tumors, an accumulation of up to 40% of EPCs mobilized from the bone marrow has been described [81].

Another important role in tumor vascularizazion is played by different cells of the tumor microenvironment, including fibroblasts, dendritic cells, macrophages, mast cells, and neutrophils, which have a crucial role in the directing the neoformation of blood vessels [82]. In this context immune cells cooperate with stromal cells as well as tumor cells in stimulating endothelial cell proliferation and blood vessel formation.

Beginning in the 1980’s, the biopharmaceutical industry began exploiting the field of anti-angiogenesis for creating new therapeutic compounds for modulating new blood vessels in tumor growth. In 2004, Avastin^®^ (bevacizumab), a humanized anti-VEGF monoclonal antibody, was the first angiogenesis inhibitor approved by the U.S. Food and Drug Administration for the treatment of colorectal cancer. However, clinical results of anti-angiogenic therapies have been very modest, resulting in a moderate improvement of overall survival, and the clinical outcome is associated with the development of resistance.

The clinical benefit of anti-angiogenic drugs is due to several intrinsic and acquired limitations including tumor indifference to anti-angiogenic therapy; selection of resistant clones and activation of alternative mechanisms that lead to activation of angiogenesis, even when the target of the drug remains inhibited.

Vasculogenic mimicry and vascular co-option are two mechanisms of both intrinsic and acquired resistance. Vasculogenic mimicry is associated with poor prognosis, reduced survival, and high risk of cancer recurrence [83]. Moreover, vasculogenic mimicry is resistant to endostatin and angiogenesis inhibitors such as TNP-470 in melanoma [84]. Post mortem histological examination of glioma bioptic specimens of patients died after receiving treatment with cediranib, an inhibitor of VEGF receptor-2 (VEGFR-2) [85], or bevacizumab [86] have shown that the glioma cells were growing around pre-existing vessels in a non-angiogenic fashion. Therapeutic strategies targeting vascular co-option might be useful to reduce resistant tumor cells.

Now is the time to transfer this knowledge from bedside to bench and then back to the bedside again. We need to understand: (a) whether non-angiogenic growth explains the limited therapeutic efficacy of anti-angiogenic therapy in patients, and (b) we need to unravel the cellular and molecular mechanisms of non-angiogenic tumors and how vessels are co-opted in cancer. Based on these insights, it may even be possible to expose new and much-needed therapeutic targets for the treatment of cancer. In this context, a new potential therapeutic approach might be represented by the combination of anti-angiogenic compounds with blockage of vascular co-option, for example, the use of bevacizumab with the antibody OS2966 against β1-integrin, one of the keys molecules in co-option [87].

## Figures and Tables

**Figure 1 cells-10-00639-f001:**
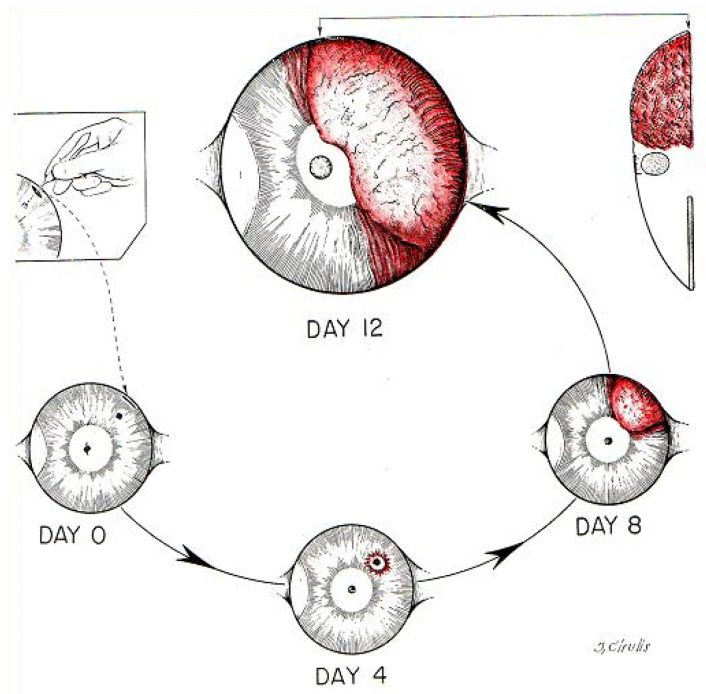
The patterns of development of two simultaneous implants of Brown-Pearce tumor in the rabbit eye. Tumors suspended in the aqueous fluid of the anterior chamber of the rabbit eye and observed for a period up to 6 weeks remain viable, avascular, of limited size (less than 1 mm^3^) and contain a population of viable and mitotically active tumor cells. After implantation contiguous to the iris, which has abundant blood vessels, the tumors induced neovascularization and grow rapidly, reaching 16,000 times the original size within two weeks. The anterior chamber implant remains avascular, while the iris implant vascularizes and grow progressively. (Reproduced from [31]).

**Figure 2 cells-10-00639-f002:**
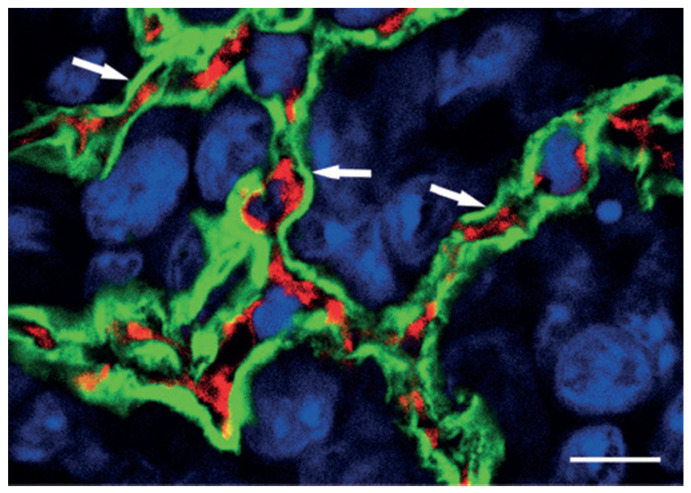
Preservation of alveolar architecture in the peripheral regions of lung metastases that is indicative of vessel co-option. Experimental lung metastases are generated by injecting HT1080 cells intravenously into SCID mice. After extravasation and forming small interstitial colonies, tumor cells enter the alveolar air spaces. This high-power confocal micrograph of the periphery of an HT1080 lung metastasis has been stained for podoplanin (green), CD31 (red), and with TOTO-3 (blue), which highlights the tumor mass. Note that intact alveolar walls with regular layering (pneumocyte–capillary–pneumocyte) are present (arrows). Scale bar = 10 μm. (Reproduced from [52]).

**Figure 3 cells-10-00639-f003:**
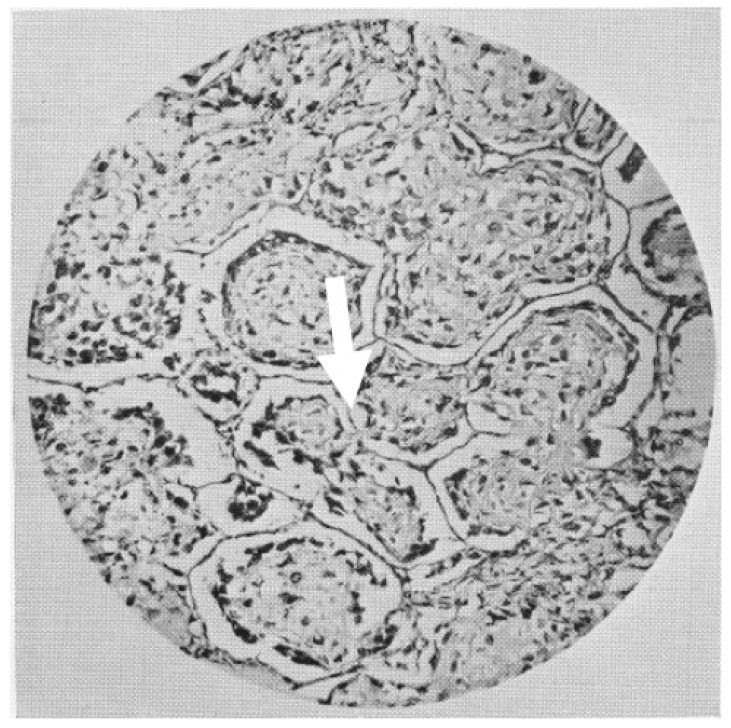
A non-angiogenic lung metastases from an osteosarcoma presented by Rupert A. Wills in his 1934 book *The Spread of the Tumours in the Human Body*. In the words of the author: “*Sections of a pulmonary metastasis showing alveoli occupied by plugs of growth resembling osteoid tissue. Note the strands of growth connecting the alveolar plugs trough the septal pores*.” (arrow).

**Figure 4 cells-10-00639-f004:**
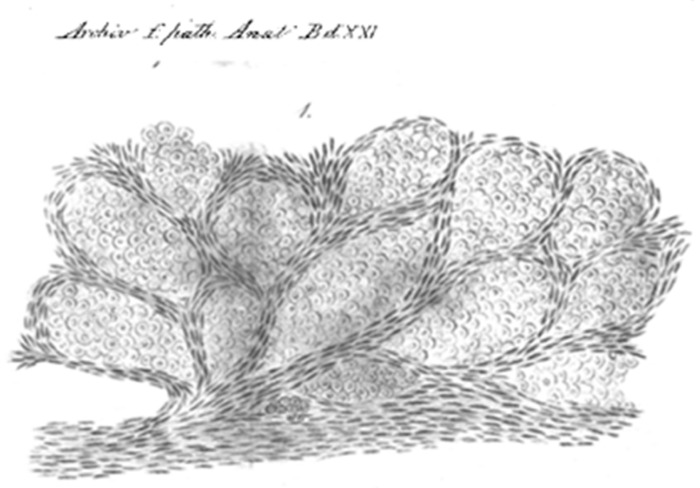
Camera Lucida drawing from a Johannes Erichsen paper of 1861 [58] is the oldest reproduction, in patients with tumors in the lung, of the neoplastic cells occupying the alveolar spaces. No new vessels can be seen while the alveolar septa containing the pre-existing capillaries, produce the “chicken wire” appearance. In the word of the author: “*Cross section through a carcinoma nodule of the lungs: the alveoli are shaped by cancerous masses confined between the elastic fibers of the lung tissue*”.

**Table 1 cells-10-00639-t001:** A brief timeline of the relationship between blood vessels and cancer.

1787	Hunter firstly used the term “angiogenesis”.
1907	Goldmann visualized angiogenesis.
1934	Detailed description of intra-alveolar growth of lung metastatic tumors spreading from one alveolus to another through the alveolar pores.
1939	Vascularization of the Brown Pearce rabbit epithelioma transplant in the transparent ear chamber in rabbits.
1956	Proliferation of cancer cells alongside hepatocellular trabeculae with a replacement pattern.
1962	Florey’s General Pathology textbook states that sometime a tumor supplement or replace the stroma using of pre-existing structures like the alveolar walls.
1971	Folkman introduces the hypothesis that tumor growth is dependent on angiogenesis.
1974	Robbins’ textbook describes that sometime metastatic cancer cells will grow in the lung without destruction of the lung parenchyma in a fashion resembling bacterial pneumonia.
1980	Demonstration that inhibition of angiogenesis suppresses tumor growth in animal models.
1988	Trabecular growth of hepatocellular carcinoma and intra-alveolar growth of primary lung carcinomas.
1996	First description of non-angiogenic tumors.
1999	First description of vascular co-option in a mouse model.
1999	First description of vasculogenic mimicry.

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
