# Peer review of "Overview on the Different Patterns of Tumor Vascularization"

_cells, 2021, doi:10.3390/cells10030639_

Round 1

Reviewer 1 Report

This is a nice historical review and it was a pleasure for me to read it. Couple of interesting historical details were new even for me in spite of many years in the field of angiogenesis.

Next to the co-option and vascular mimicry another alternative way of tumor angiogenesis, so called intussusceptive angiogenesis has been reported. Even more, it seems that this complementary way of blood vessel formation could be switched one during tumor treatment. Both irradiation and anti-angiogenic therapy cause a switch from sprouting to intussusceptive angiogenesis, representing an escape mechanism and accounting for the development of resistance, as well as rapid recovery, after cessation of therapy. It will be very informative, if the authors will include a small paragraph dealing with this topic.

Author Response

Next to the co-option and vascular mimicry another alternative way of tumor angiogenesis, so called intussusceptive angiogenesis has been reported. Even more, it seems that this complementary way of blood vessel formation could be switched one during tumor treatment. Both irradiation and anti-angiogenic therapy cause a switch from sprouting to intussusceptive angiogenesis, representing an escape mechanism and accounting for the development of resistance, as well as rapid recovery, after cessation of therapy. It will be very informative, if the authors will include a small paragraph dealing with this topic.

We have added a new paragraph concerning intussusceptive angiogenesis as follows: “Next to the co-option and vascular mimicry another alternative way of tumor angiogenesis, so called intussusceptive angiogenesis or intussusceptive microvascular growth (IMG) has been reported. IMG is characterized by the insertion of newly formed columns of interstitial tissue structures called tissue pillars into the vascular lumen [79]. Even more, it seems that this complementary way of blood vessel formation could be switched one during tumor treatment with various antitumor and anti-angiogenic molecules. Both irradiation and anti-angiogenic therapy cause a switch from sprouting to IMG, representing an escape mechanism and accounting for the development of resistance, as well as rapid recovery, after cessation of therapy [80].

Reviewer 2 Report

This review has a narrative style and retraces, from a historical point of view, the main stages of the studies carried out on tumor angiogenesis. It also focuses on the description of the main studies that have contributed to the knowledge of this process.

However, the referee feels to point out some aspects that represent a weakness of the manuscript such as:

  1. The authors state: “The effect of microvessel density as prognostic indicator has been however questioned by a meta-analysis on original data that failed to confirm the value of microvessel density as a prognostic marker in non-small cell lung carcinoma [45]”. The concept of microvessel density (MVD) should perhaps be expanded to include the concept of the inter-capillary distance. The review would benefit from this, also because it would clarify to the reader some obscure points relating to antiangiogenic treatments in anticancer therapy. In fact, MVD is a feature of several cancers and not just lung cancers. Furthermore, many benign tumors (e.g. adrenal adenoma) have a higher microvessel density than some malignant tumors (e.g. chondrosarcoma).

  1. In the statement: “More recent studies have shown that this replacement growth pattern of liver metastases is indeed non-angiogenic Vermeulen et al.[49], as the cancer cells co-opt the sinusoidal vasculature of the liver”. Perhaps “Vermeulen et al.” should be omitted from the text.

  1. “The basement membrane stained with periodic acid Schiff (PAS) can be observed in these vascular channels, but endothelial cells were not identified [75]. Microarray gene chip analysis of a highly aggressive compared with poorly aggressive human cutaneous melanoma cell lines revealed a significant increase in the expression of laminin 5 and matrix metalloproteinases-1, -2, and -9 (MMP-1, MMP-2, MMP-9) and MT1-MMP in the highly aggressive cells [76], suggesting that increased expression of MMP-2 and MT1-MMP along with matrix deposition of laminin 5 are required for their mimicry”. Although the authors describe the molecular mechanisms associated with vasculogenic mimicry, a more detailed description of the process would be of great help to readers inexperienced in tumor angiogenesis. The authors cite Ref. 74 (by Maniotis et al.). This is the first study demonstrating the high plasticity of cancer cells to participate in the construction of new vessels in response to increased hypoxia. Indeed, the transendothelial differentiation of melanoma cells results in the formation of a chimeric vasculature composed of human melanoma and mouse endothelial cells. After restoring blood flow, the melanoma cells form a large tumor mass.

  1. The authors do not mention the participation of endothelial precursor cells (EPC) in the construction of the new vessels, namely adult vasculogenesis. It is known that the angiogenic switch can promote adult vasculogenesis by enhancing EPC recruitment and vessel formation at the site of tumor neovascularization. They should add this process after paragraph 6. Furthermore, nowhere in the manuscript is there any reference to hypoxia as a trigger for the construction of new vessels and for the selection of more invasive clones, particularly in the introduction.

  1. Since the manuscript was presented in the special issue "Non-angiogenic tumors: from cellular mechanisms to new therapeutic approaches" perhaps a hint on the dialogue between angiogenic factors, the tumor microenvironment and the blunted immune response in the conclusions could better fit the manuscript into this special issue.

Author Response

However, the referee feels to point out some aspects that represent a weakness of the manuscript such as:

The authors state: “The effect of microvessel density as prognostic indicator has been however questioned by a meta-analysis on original data that failed to confirm the value of microvessel density as a prognostic marker in non-small cell lung carcinoma [45]”. The concept of microvessel density (MVD) should perhaps be expanded to include the concept of the inter-capillary distance. The review would benefit from this, also because it would clarify to the reader some obscure points relating to antiangiogenic treatments in anticancer therapy. In fact, MVD is a feature of several cancers and not just lung cancers. Furthermore, many benign tumors (e.g. adrenal adenoma) have a higher microvessel density than some malignant tumors (e.g. chondrosarcoma).

We have expanded the concept of MVD including the concept of intercapillary distance as follows:

 “It is important to consider the relationship between MVD and intercapillary distance. Intercapillary distance is determined at the local level by the net balance between angiogenic and anti-angiogenic factors, as well as by the oxygen and nutrient consumption rates of tumor cells [46]. In turn, MVD is determined by intercapillary distance, which in a tumor is dictated by the thickness of the perivascular cuff of tumor cells. Tumors that have high rates of oxygen and nutrient consumption have small cuffs only two to three cells wide and have high vascular density [47]. On the contrary, tumors with low rates of oxygen consumption have relatively large cuff sizes and a relatively low vascular density [47].This is an important parameter as the goal of an anti-angiogenic tumor therapy to reduce the intercapillary distance to a such a degree that it becomes rate-limiting for the tumor growth.”

In the statement: “More recent studies have shown that this replacement growth pattern of liver metastases is indeed non-angiogenic Vermeulen et al.[49], as the cancer cells co-opt the sinusoidal vasculature of the liver”. Perhaps “Vermeulen et al.” should be omitted from the text.

DONE 

“The basement membrane stained with periodic acid Schiff (PAS) can be observed in these vascular channels, but endothelial cells were not identified [75]. Microarray gene chip analysis of a highly aggressive compared with poorly aggressive human cutaneous melanoma cell lines revealed a significant increase in the expression of laminin 5 and matrix metalloproteinases-1, -2, and -9 (MMP-1, MMP-2, MMP-9) and MT1-MMP in the highly aggressive cells [76], suggesting that increased expression of MMP-2 and MT1-MMP along with matrix deposition of laminin 5 are required for their mimicry”. Although the authors describe the molecular mechanisms associated with vasculogenic mimicry, a more detailed description of the process would be of great help to readers inexperienced in tumor angiogenesis. The authors cite Ref. 74 (by Maniotis et al.). This is the first study demonstrating the high plasticity of cancer cells to participate in the construction of new vessels in response to increased hypoxia. Indeed, the transendothelial differentiation of melanoma cells results in the formation of a chimeric vasculature composed of human melanoma and mouse endothelial cells. After restoring blood flow, the melanoma cells form a large tumor mass.

We have improved the description of vasculogenic mimicry as follows: “This is the first study demonstrating the high plasticity of cancer cells to participate in the construction of new vessels in response to increased hypoxia. Indeed, the transendothelial differentiation of melanoma cells results in the formation of a chimeric vasculature composed of human melanoma and mouse endothelial cells. After restoring blood flow, the melanoma cells form a large tumor mass. This process occurs in the absence of endothelial cells and the tumor cells mimic the vascular endothelium”.

Since the manuscript was presented in the special issue "Non-angiogenic tumors: from cellular mechanisms to new therapeutic approaches" perhaps a hint on the dialogue between angiogenic factors, the tumor microenvironment and the blunted immune response in the conclusions could better fit the manuscript into this special issue.

We have added a paragraph concerning the role of tumor microenvironment as follows: “Another important role in tumor vascularizazion is played by different cells of the tumor microenvironment, including fibroblasts, dendritic cells, macrophages, mast cells, and neutrophils, which have a crucial role in the directing the neoformation of blood vessels [84]. In this context immune cells cooperate with stromal cells as well as tumor cells in stimulating endothelial cell proliferation and blood vessel formation.”

Reviewer 3 Report

In general, this review is very poor written. The sections must be largely improved, as a lot of data is missing. For instance, in the section 4. Tumor angiogenesis in human tumor progression and as a prognostic indicator, no general of molecular mechanisms of angiogenesis were provided. The same for sections 6 and 7, 6. Vascular co-option and 7. Vasculogenic mimicry. In my opinion, the review must be rewritten, reorganizwed and be rich in historical info, as in its actual form did not reach the high standar for Cells journal.

Abstract section: The aim of this review os lacking, please describe the rationale of this manuscript.

Figure 1 and 2 must be removed (WIKIPIDEA source is unreliable). 

Make a more detailed description of the patterns of development of two simultaneous implants of Brown-Pearce tumor in the rabbit eye, both in the manuscript and in the figure, as it is very general.  

In figure 4, 5 and 6 the components within the figure must be fully explained.

It is suggested to add a table of contents or timeline of description of points 2 to 7.  

The form quoted in the figure caption is not adequate.  

Standardize the format of bibliographic references.  

The conclusion is read as a summary, point out / highlight the contribution of the manuscript

Author Response

In general, this review is very poor written. The sections must be largely improved, as a lot of data is missing. For instance, in the section 4. Tumor angiogenesis in human tumor progression and as a prognostic indicator, no general of molecular mechanisms of angiogenesis were provided. The same for sections 6 and 76. Vascular co-option and 7. Vasculogenic mimicry. In my opinion, the review must be rewritten, reorganized and be rich in historical info, as in its actual form did not reach the high standard for Cells journal.

We have improved all the sections of the ms. In detail, we have provided general molecular mechanisms of angiogenesis as follows: “Angiogenesis and production of angiogenic factors are fundamental for tumor progression in form of growth, invasion, and metastasis. New vessels promote growth by conveying oxygen and nutrients, and removing catabolites. In some tumors, an oncogene is responsible for overexpression of a positive regulator of angiogenesis. Ras oncogene upregulates expression of VEGF. In other tumor types, concomitant down-regulation of an endogenous negative regulator of angiogenesis is also required for the angiogenic switch. When the tumor suppressor gene TP53 is mutated or deleted, thrombospondin-1 (TSP-1), an endogenous angiogenesis inhibitor is downregulated. Tumor angiogenesis can be potentiated by hypoxia, which activates a hypoxia-inducible factor (HIF-1)-binding sequence in the VEGF promoter, leading to transcription of VEGF mRNA, and increased production of VEGF protein. HIF-1 also increases the transcription of genes for platelet-derived endothelial growth factor (PDGF)-BB and nitric oxide synthetase (NOS). Tumor angiogenesis can also be potentiated by angiogenic proteins released by inflammatory cells present in the tumor microenvironment.”

Moreover, we have described the tole of postnatal vasculologenesis, of EPCs, and of the cells of tumor microenvironment as follows: “Almost two other mechanisms together those previously described in this article are involved in the formation of tumor vessels, i.e. post-natal tumor vasculogenesis and endothelial prercursor cells (EPCs) activation  [81,82]. Vasculogenesis may not be restricted to early embryogenesis, but may also contribute to tumor vascularization in postnatal life. This comes from observations that both endothelial cells and EPCs co-exist in the circulation and that blood vessels can emerge in an alternative way during tumor growth. EPCs may derive from the bone marrow and not yet be incorporated into the vessel wall. Several evidences suggest that EPCs constitute the preponderance of circulating bone marrow-derived endothelial lineage cells. In some tumors, an accumulation of up to 40% of EPCs mobilized from the bone marrow has been described [83]. Another important role in tumor vascularizazion is played by different cells of the tumor microenvironment, including fibroblasts, dendritic cells, macrophages, mast cells, and neutrophils, which have a crucial role in the directing the neoformation of blood vessels [84]. In this context immune cells cooperate with stromal cells as well as tumor cells in stimulating endothelial cell proliferation and blood vessel formation.”

Abstract section: The aim of this review is lacking, please describe the rationale of this manuscript.

We have improved the Abstract as follows: In this article, we have analyzed the mechanisms involved in tumor vascularization in association with classical angiogenesis, including post-natal vasculogenesis, intussusceptive microvascular growth, vascular co-option, and vasculogenic mimicry. We have also discussed the role of these alternative mechanism in resistance to anti-angiogenic therapy and potential therapeutic approaches to overcome resistance.”

Figure 1 and 2 must be removed (WIKIPIDEA source is unreliable). 

DONE

Make a more detailed description of the patterns of development of two simultaneous implants of Brown-Pearce tumor in the rabbit eye, both in the manuscript and in the figure, as it is very general.

We have improved the description as follows: Folkman and collaborators provide further evidence for the dependence of tumor growth on neovascularization: tumor growth in the avascular cornea proceeds slowly at a linear rate, but after vascularization, tumor growth is exponential [32,33]. Tumors suspended in the aqueous fluid of the anterior chamber of the rabbit eye and observed for a period up to 6 weeks remain viable, avascular, of limited size (less than 1 mm3) and contain a population of viable and mitotically active tumor cells.  After implantation contiguous to the iris, which had abundant blood vessels, the tumors induced neovascularization and grow rapidly, reaching 16,000 times the original size within two weeks After one week, new blood vessels had invaded the cornea starting from the edge closer to the site of implantation and developed in that direction at 0.2 mm and then about 1 mm/day Once the vessels reached the tumor, it grew very rapidly to permeate the entire globe within four weeks. (Figure 1). “ 

In figure 4, 5 and 6 the components within the figure must be fully explained.

We have improved the description of the figures (now Figures 2, 3 and 4).

It is suggested to add a table of contents or timeline of description of points 2 to 7.  

WE HAVE ADDED A TIMELINE reported in the Table 1.

The form quoted in the figure caption is not adequate.  

Standardize the format of bibliographic references.

DONE  

The conclusion is read as a summary, point out / highlight the contribution of the manuscript.

We have re-written the last section as follows: “Almost two other mechanisms together those previously described in this article are involved in the formation of tumor vessels, i.e. post-natal tumor vasculogenesis and endothelial prercursor cells (EPCs) activation  [81,82]. Vasculogenesis may not be restricted to early embryogenesis, but may also contribute to tumor vascularization in postnatal life. This comes from observations that both endothelial cells and EPCs co-exist in the circulation and that blood vessels can emerge in an alternative way during tumor growth. EPCs may derive from the bone marrow and not yet be incorporated into the vessel wall. Several evidences suggest that EPCs constitute the preponderance of circulating bone marrow-derived endothelial lineage cells. In some tumors, an accumulation of up to 40% of EPCs mobilized from the bone marrow has been described [83].

Another important role in tumor vascularizazion is played by different cells of the tumor microenvironment, including fibroblasts, dendritic cells, macrophages, mast cells, and neutrophils, which have a crucial role in the directing the neoformation of blood vessels [84]. In this context immune cells cooperate with stromal cells as well as tumor cells in stimulating endothelial cell proliferation and blood vessel formation.

Beginning in the 1980’s, the biopharmaceutical industry began exploiting the field of anti-angiogenesis for creating new therapeutic compounds for modulating new blood vessels in tumor growth. In 2004, Avastin (Bevacizumab), a humanized anti-VEGF monoclonal antibody, was the first angiogenesis inhibitor approved by the Food and Drug Administration for the treatment of colorectal cancer. However, clinical results of anti-angiogenic therapies have been very modest, resulting in a moderate improvement of overall survival, and the clinical outcome is associated with the development of resistance.

The clinical benefit of anti-angiogenic drugs is due to several intrinsic and acquired limitations including tumor indifference to anti-angiogenic therapy; selection of resistant clones and activation of alternative mechanisms that lead to activation of angiogenesis, even when the target of the drug remains inhibited.

Vasculogenic mimicry and vascular co-option are two mechanisms of both intrinsic and acquired resistance.  Vasculogenic mimicry is associated with poor prognosis, reduced survival, and high risk of cancer recurrence [85]. Moreover, vasculogenic mimicry is resistant to endostatin and angiogenesis inhibitors such as TNP-470 in melanoma [86]. Post mortem histological examination of   glioma bioptic specimens of patients died after receiving treatment with Cediranib, an inhibitor of VEGFR-2 tyrosine kinases [87], or bevacizumab [88] have shown that the glioma cells were growing around pre-existing vessels in a non-angiogenic fashion. Therapeutic strategies targeting vascular co-option might be useful to reduce resistant tumor cells.

Now is the time to transfer this knowledge from bedside to bench and then back to the bedside again. We need to understand: (a) whether non-angiogenic growth explains the limited therapeutic efficacy of anti-angiogenic therapy in patients, and (b) we need to unravel the cellular and molecular mechanisms of non-angiogenic tumors and how vessels are co-opted in cancer. Based on these insights, it may even be possible to expose new and much-needed therapeutic targets for the treatment of cancer. In this context, a new potential therapeutic approach might be represented by the combination of anti-angiogenic compounds with blockage of vascular co-option, for example, the use of bevacizumab with the antibody OS2966 against β1integrin, one of the keys molecules in co-option [89].”

Round 2

Reviewer 3 Report

Authors have improved the minireview.